# Association of TNF–α rs1800629 with Adult Acute B-Cell Lymphoblastic Leukemia

**DOI:** 10.3390/genes13071237

**Published:** 2022-07-13

**Authors:** Ezeldine K. Abdalhabib, Abdulrahman Algarni, Muhammad Saboor, Fehaid Alanazi, Ibrahim K. Ibrahim, Ayman H. Alfeel, Abdullah M. Alanazi, Abdulmajeed M. Alanazi, Abdulaziz M. Alruwaili, Muath H. Alanazi, Nahla A. Alshaikh

**Affiliations:** 1Department of Clinical Laboratory Sciences, College of Applied Medical Sciences, Jouf University, Sakakah P.O. Box 42421, Saudi Arabia; ezeldine@ju.edu.sa (E.K.A.); fmaalanazi@ju.edu.sa (F.A.); 2Department of Medical Laboratory Technology, College of Applied Medical Sciences, Northern Border University, Arar P.O. Box 91431, Saudi Arabia; abdulrahman.eid@nbu.edu.sa; 3Department of Medical Laboratory Technology, Faculty of Applied Medical Sciences, Jazan University, Jazan P.O. Box 45142, Saudi Arabia; nalshaikh@jazanu.edu.sa; 4Medical Research Center, Jazan University, Jazan P.O. Box 45142, Saudi Arabia; 5Department of Hematology, Faculty of Medical Laboratory Sciences, Al Neelain University, Khartoum P.O. Box 12702, Sudan; ibrahimkh82@gmail.com; 6Department of Medical Laboratory Sciences, College of Health Sciences, Gulf Medical University, Ajman P.O. Box 4184, United Arab Emirates; 7Gurayyat Health Affair, Regional Laboratory and Central Blood Bank, Ministry of Health, Gurayyat P.O. Box 77413, Saudi Arabia; aalanazi29@moh.gov.sa (A.M.A.); goda111@hotmail.com (A.M.A.); aalruwaili77@moh.gov.sa (A.M.A.); muathha@moh.gov.sa (M.H.A.)

**Keywords:** acute lymphoblastic leukemia, rs1800629, polymorphism, TNF–α

## Abstract

TNF–α influences lymphomagenesis by upregulating proinflammatory and antiapoptotic pathways. In this study, we evaluated the frequency of TNF–α rs1800629 (–308 G>A) polymorphism in newly diagnosed adult patients with acute lymphoblastic leukemia (ALL) and its correlation with age at diagnosis, gender and subtype of ALL. In this case control study, a total of 330 individuals were recruited, including 165 newly diagnosed adult patients with ALL, from the Radiation and Isotope Center in Khartoum (RICK) and 165 healthy normal controls. TNF–α rs1800629 polymorphism was tested through allele-specific polymerase chain reaction (PCR) assay. The frequency of the rs1800629 GA genotype was high (70.9% vs. 60%, OR = 1.84) in the patient group as compared to healthy controls, whereas GG and AA genotypes did not exhibit any statistically significant difference between controls and patients. Based on subtype, GG and GA rs1800629 genotypes showed increased risk of B-ALL (OR 0.46 and 2.12, respectively), whereas rs1800629 GG, GA and AA genotypes did not show any disease association with T-ALL (*p* > 0.05). Age at diagnosis and gender did not exhibit any association of rs1800629 with ALL in the patient group. In conclusion, rs1800629 is associated with high risk of adult B-ALL, with an insignificant effect of age at diagnosis and gender.

## 1. Introduction

Acute lymphoblastic leukemia (ALL) is a lymphoid neoplasm that results in uncontrolled and restricted neoplastic proliferation of lymphoblasts in bone marrow. Over time, these lymphoblasts flood the peripheral blood and infiltrate extramedullary sites. It is more common in children, comprising more than 80% of all cases. Although rare, ALL exhibits severe clinical manifestations in adult patients [1,2]. Although pathological findings, diagnostic features, immunophenotypic characteristics and cytogenetic aberrations have been elucidated, the exact genetic role in the pathophysiology of ALL has not yet been well-defined yet [3]. Inherited human genetic disorders, e.g., Down syndrome, Fanconi’s anemia, Bloom syndrome, ataxia telangiectasia and Nijmegen breakdown syndrome, have been found to be significant risk factors for ALL. Several genetic aberrations affecting oncogenes and tumor suppresser genes, such as neuroblastoma RAS viral (v–ras) oncogene homolog (NRAS), p53 and PHD finger protein (PHF6), have been detected in many cases of ALL. Additionally, cytogenetic anomalies and several environmental factors pose a risk for development of ALL [1].

Tissue necrotic factor–α (TNF–α), a pleiotropic cytokine of the tumor necrosis family, is amongst the pivotal regulators of inflammation, immunity, apoptosis and hematopoiesis [4]. TNF–α is produced by macrophages, monocytes, neutrophils, T cells and natural killer cells [5]. It is produced in inactive, membrane-bound form, with a molecular weight of 26 kDa, which is converted by a metalloproteinase disintegrin called TNF-α converting enzyme (TACE) into 17 kDa active soluble form. Additionally, TACE regulates the concentration of soluble tissue necrotic factor receptor 2 (TNFR2) by acting on membrane-bound TNFR2 protein [6]. TNFR1, ubiquitously expressed in almost all types of cells, interacts with sTNF and induces a proinflammatory response, whereas the binding of mTNF to TNFR2 triggers immune modulation [7]. TNF–α plays two different roles, depending upon the stimulus and local or systemic production. When produced locally, TNF–α enhances the egress of inflammatory cells, i.e., neutrophils and monocytes, by increasing the expression of adhesion molecules and activates phagocytosis of infectious pathogens and cell debris by these cells. Conversely, TNF–α may have detrimental effects, causing fever and increased levels of glucocorticoids by inducing the synthesis of IL–1 and IL–6 [6].

TNF–α, with the help of nuclear transcription factor kappa B (NF–κB) pathway, is thought to influence lymphomagenesis by upregulating proinflammatory and antiapoptotic pathways. NF–κB stimulates the production of proinflammatory cytokines, which leads to proliferation and survival of lymphoid cells. Secondly, antiapoptotic properties of NF–κB inhibit apoptosis among cells with neoplastic potential [8,9]. Any imbalance in this process could lead to enhanced lymphomagenesis. Hence, TNF–α may be a link between neoplasia and inflammation by promoting cellular transformation, enhanced proliferation, prolonged survival, invasion and neoangiogenesis [10].

TNF–α is encoded by the *TNF–α* gene located on chromosome 6, in close proximity to a highly polymorphic region of the class III HLA–B locus of human major histocompatibility complex [6,11]. It is a well-known fact that the gene products are affected by polymorphic markers in this region. A single-nucleotide polymorphism (SNP) of TNF–α promotor gene results in two types of alleles, i.e., TNF1 and TNF2. TNF1 is a common allele defined by the presence of guanine, whereas TNF2 is a rarer allele characterized by the substitution of guanine with adenosine, located at position –308 (NC_000006.11:g.31543031 G>A, rs1800629), which influences the expression of TNF–α [6,10,12,13].

The role of TNF–α rs1800629 polymorphism has been evaluated under various clinical conditions, e.g., autoimmune disorders (systemic lupus erythematosus, rheumatoid arthritis and ankylosing spondylitis), inflammatory diseases (Celiac disease, Crohn’s disease, meningococcal disease, acute pancreatitis, asthma and sepsis), coronary artery diseases, type 2 diabetes mellitus and in patients with transplants [6,14,15]. The literature contains contradictory reports regarding the frequency of rs1800629 TNF–α promotor gene polymorphism and its association with leukemia. Significantly increased frequency of rs1800629 has been reported in some studies in leukemic patients [16,17], whereas no association was reported in other studies [4,18].

Keeping in view the role of TNF–α in the proposed lymphomagenesis, SNP in the TNF–α promotor gene could be associated with lymphoproliferative disorders. Several studies have evaluated the frequency of rs1800629 in lymphoid neoplasms, including non-Hodgkin lymphoma (NHL), chronic lymphocytic leukemia and childhood acute lymphoblastic leukemia, with contradictory reports [14,17,18,19,20]. Due to scant data addressing the association of adult ALL and TNF–α rs1800629 polymorphism, this study was designed to evaluate the frequency of TNF–α rs1800629 (–308 G>A) in newly diagnosed adult patients with ALL and its correlation with age at diagnosis, gender and subtype of ALL.

## 2. Materials and Methods

In this case control study, a total of 330 individuals were recruited, including 165 newly diagnosed adult patients with ALL (101 male and 64 female) and 165 healthy normal controls (107 male and 58 female). Age of the participants was in the range of 19–75 years. The newly diagnosed ALL patient group was enrolled prior to initiation of the treatment for ALL at the Radiation and Isotope Center in Khartoum (RICK), Sudan. This study was conducted between December 2019 and April 2022 according to the Declaration of Helsinki after receiving approval from the Al–Neelian University ethical committee. Written informed consent was obtained from each study participant prior to sample collection. Diagnosis of ALL was based on the findings of complete blood counts, bone marrow examination and flow cytometric analysis by a hemato-oncologist. To represent the same age group and ethnicity, a control group was selected from the same geographical area, consisting of subjects without any history of cancer or hematological neoplasms. Demographic data and clinical features were collected using a well-structured questionnaire. An EDTA–anticoagulated blood sample was collected from each study participant for polymerase chain reaction analysis.

### 2.1. DNA Extraction

The EDTA–anticoagulated blood samples were used for the isolation of DNA with a whole-blood genomic DNA purification kit (QIAamp^®^ DNA Mini kit; Qiagen GmbH, Hilden, Germany) as per the manufacturer’s instructions. A gene quant device (Amersham bioscience–Biochrome LTD, Cambridge CB4, England) was used to analyze the quantity and quality of the extracted DNA. Aliquots of the extracted DNA were stored at −20 °C until analysis.

### 2.2. Molecular Analysis

The aliquots stored at −20 °C were thawed and used for the genotyping of TNF–α rs1800629 polymorphism through allele-specific polymerase chain reaction (PCR) assay using β–globin gene-amplified product as an internal positive control following a previously described method [21]. For the TNF–α rs1800629 polymorphism, the following three primers were used: 5′-ATAGGTTTTGAGGGGCATGG-3′ (allele G forward), 5′-AATAGGTTTTGAGGGGCATGA-3′ (allele A forward) and 5′-CAGCCCTTCCATTTTACTTTC-3′ (reverse). The thermocycling conditions started with one cycle at 94 °C for 5 min; 32 cycles at 94 °C for 30 s, annealing at 58 °C for 30 s and extension at 72 °C for 40 s; with a final extension at 72 °C for 5 min. The amplified products were electrophoresed on 2% agarose gel to examine the banding patterns with the help of ethidium bromide. A PCR product with 268 bp indicated successful amplification. Based on the presence or absence of a band at 184 bp, TNF–α rs1800629 (Figure 1).

### 2.3. Statistical Analysis

Statistical analyses, including descriptive statistics of mean, standard deviation and odds ratio (OR) with a confidence interval (CI) of 95%, were performed using Statistical Package For Social Sciences (SPSS) for Windows (Chicago, IL, USA), version 23. Pearson chi-square test or Fisher’s exact test was used to evaluate the significance of differences in genotype distributions of patients and controls. Quantitative variables were also tested using an independent *t*-test. Statistical significance was judged by a *p* value less than 0.05.

## 3. Results

The patient cohort of this study consisted of a total of 165 patients, including 61.2% (n = 101) males and 38.8% (n = 64) females with newly diagnosed ALL. The normal control group comprised 64.8% males and 35.2% females with no statistical difference (*p* > 0.05). Mean age of the patients was 44.64 ± 14.28, and that of the control group was 46.72 ± 15.87 (*p* > 0.05). A total of 83% patients had B-ALL, whereas 17% had T-ALL (Table 1).

The frequency of rs1800629 GA genotype was high (70.9% vs. 60%, OR = 1.84) in the patient group as compared to healthy controls, whereas GG and AA genotypes did not exhibit any statistically significant difference between controls and patients. Combination analysis of GA + AA vs. GG and GA + GG vs. AA genotypes of rs1800629 also showed a statistically insignificant difference (*p* > 0.05). Similarly, G or A alleles were equally distributed among adult patients with ALL and the control group (*p* > 0.05) (Table 2).

GG and GA rs1800629 genotypes were associated with increased risk of B–ALL with OR of 0.46 and 2.12, respectively (Table 3) whereas rs1800629 GG, GA and AA genotypes did not show any disease association with T-ALL (*p* > 0.05). Age at diagnosis, gender and immunophenotype (B versus T-ALL) did not exhibit any association of rs1800629 with ALL in patient group (Table 4).

## 4. Discussion

In addition to its most vital and well-recognized role as a proinflammatory cytokine, TNF–α also participates in the regulation and enhanced proliferation of inflammatory cells, as well as apoptosis. Hence, it could be a risk factor for the development of hematological neoplasias and their outcomes.

In the current study, the GA genotype of the TNF–α gene revealed high frequency, with a 1.84-fold increased risk of ALL in adult patients as compared to normal control subjects, which is in accordance with the results of a study carried out on a pediatric group [17]. In another study in which rs1800629 was analyzed in children and adult patients with ALL, a high frequency of mutant GA was found in the pediatric group but not in adults [22]. Furthermore, in the current study, GG/AA genotypes and G/A alleles did not exhibit any association with adult ALL, which is in contrast to the findings of AbdEl-Aziz et al., who found these genotypes and alleles to be a significant risk factor of childhood ALL [22]. In a meta–analysis, although increased frequency of the AA genotype was reported in patients with ALL, it was statistically insignificant [4]. Similar to the findings of the meta-analysis, the current study also did not reveal any association of the AA genotype with adult ALL. Additionally, GG and GA genotypes were highly prevalent in B-ALL (OR = 0.46 and 2.12 respectively). According to combination analysis, GA + AA vs. GG genotype and GA + GG vs. AA did not pose any significant risk. In a meta-analysis of 19 relevant studies, no association of rs1800629 was found to be a risk factor of leukemia. Furthermore, in support of the findings of the current study, no association of the GG genotype, A allele, G allele and AA vs. GG genotypes and risk of leukemia including ALL and chronic lymphocytic leukemia was reported by Gong et al. [4]. Moreover, in the current study, no association of rs1800629 with age or gender was noted. These differences could be related to genetic variation among populations, environmental factors, course and severity of the disease, as well as a small number of study participants in the referenced studies.

It has been documented that the A allele of the altered TNF–α promotor region due to rs1800629 SNP leads to higher transcriptional activity than its ancestral G allele, causing increased production of TNF–α by B and T cells [23,24,25,26]. High levels of TNF–α have been found in patients with childhood ALL in comparison to normal control subjects [27]. This raises concerns as to whether high plasma levels of TNF–α are associated with neoplasia or inflammatory response towards malignancy. High concentrations of TNF–α in ALL patients were found to be correlated with significantly increased peripheral blood blast count [27,28]. However, the blast count decreased significantly during the therapeutic induction phase due the inhibitory effect of glucocorticoids on TNF–α expression [27]. In patients with NHL, high levels of TNF–α were been found to be associated with negative prognosis, high frequency of relapse and reduced survival [10]. Similarly, it was reported that the G>A genotype and A allele of rs1800629 SNP increase susceptibility to NHL and autoimmune rheumatic disease with lymphoproliferative disorders [14,23].

## 5. Conclusions

In conclusion, rs1800629 is associated with high risk of adult B-ALL, with an insignificant effect of age at diagnosis and gender. The GA genotype of the TNF–α gene is a risk factor for ALL in adult patients. Additionally, GG and GA genotypes were highly prevalent in B-ALL (OR = 0.46 and 2.12 respectively). GG/AA genotypes and G/A alleles are not associated with adult ALL T–ALL. According to combination analysis, GA + AA vs. GG genotype and GA + GG vs. AA do not pose any significant risk. It is highly recommended to carry out large-scale studies to explore the pathophysiological role of rs1800629 in ALL and its management.

## Figures and Tables

**Figure 1 genes-13-01237-f001:**
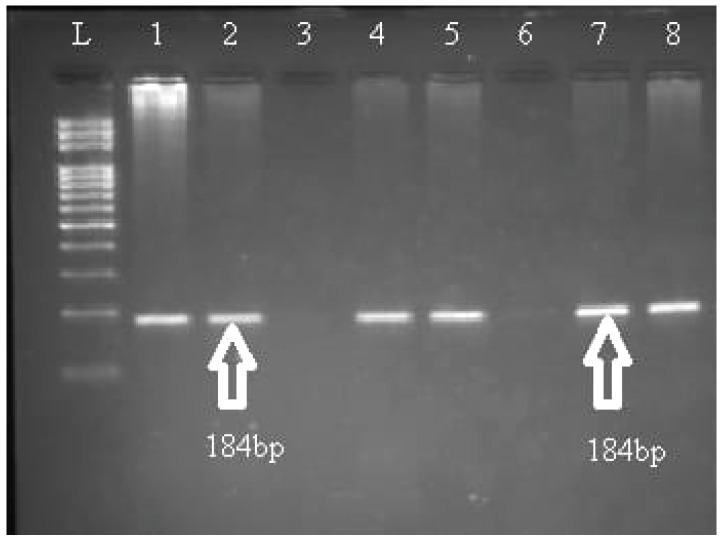
TNF-α genotyping using AS-PCR. Lane L: 100 bp DNA molecular weight marker. Lanes 1, 5 and 7: allele G is represented by the presence of a 184 bp PCR fragment. Lanes 2, 4 and 8: allele G is represented by the presence of a 184 bp PCR fragment. Each set of two lanes represents one sample: lanes 1 and 2, heterozygous (GA); lanes 3 and 4, homozygous (AA); lanes 5 and 6, homozygous (GG); lanes 7 and 8, heterozygous (GA).

**Table 1 genes-13-01237-t001:** Demographic measures of the studied participants.

Category	Cases (n = 165)	Controls (n = 165)	(*p*-Value)
**Gender**			
**Male n (%)**	101 (61.2%)	107(64.8%)	0.494
**Female n (%)**	64 (38.8%)	58 (35.2%)	
**Age**			
**Mean ± SD (year)**	44.64 ± 14.28	46.72 ± 15.87	0.213
**Range (year)**	19–75	18–74	
**<mean age n (%)**	84 (50.9%)	92 (55.8%)	0.377
**>mean age n (%)**	81 (49.1%)	73 (44.2%)	
**Immunophenotype**			
**B-ALL n (%)**	137 (83%)	-	-
**T-ALL n (%)**	28 (17%)	-	-

**Table 2 genes-13-01237-t002:** The genotype distribution and allele frequencies of TNF rs1800629 in the ALL cases and the control group.

SNP	Genotype	ALL (n = 165)	Controls (n = 165)	Odds Ratio (95% CI)	*p*-Value
**rs1800629**	GG	25 (15.2%)	39 (23.6%)	0.58 (0.33 to 1.01)	0.053
	GA	117 (70.9%)	94 (60%)	1.84 (1.17 to 2.90)	0.009
	AA	23 (13.9%)	32 (19.4%)	0.67 (0.37 to 1.21)	0.185
	GA +AA vs. GG	140 (84.8%)	126 (76.4%)	1.73 (0.99 to 3.03)	0.053
	GA +GG vs. AA	142 (86.1%)	134 (81.2%)	1.47 (0.82 to 2.65)	0.194
**Allele frequency**					
	G	167 (50.6%)	172 (52.1%)	0.94 (0.69 to 1.28)	0.697
	A	163 (49.4%)	158 (47.9%)		

**Table 3 genes-13-01237-t003:** Genotype distribution of rs1800629 in the B–ALL/T–ALL cases and control group.

SNP	Genotype	B-ALL (n = 137)	Controls (n = 165)	Odds Ratio (95% CI)	*p*-Value	T-ALL (n = 28)	Controls (n = 165)	Odds Ratio (95% CI)	*p*-Value
**rs1800629**	GG	17 (12.4%)	39 (23.6%)	0.46 (0.25 to 0.85)	0.014	8 (28.6%)	39 (23.6%)	1.29 (0.53 to 3.16)	0.575
	GA	101 (73.7%)	94 (60%)	2.12 (1.30 to 3.46)	0.003	16 (57.1%)	94 (60%)	1.01 (0.45 to 2.26)	0.986
	AA	19 (13.9%)	32 (19.4%)	0.67 (0.36 to 1.24)	0.204	4 (14.3%)	32 (19.4%)	0.69 (0.22 to 2.14)	0.523

**Table 4 genes-13-01237-t004:** Distribution of rs1800629 among patients according to age group, gender and origin of ALL.

Parameter	Category	rs1800629		
		GG	GA	AA
**Age group**	**<mean years**	11	63	10
	**>mean years**	14	54	13
**Chi square (*p*-value)**		1.39 (0.499)		
**Gender**	**Male**	16	69	16
	**Female**	9	48	7
**Chi square (*p*-value)**		1.0 (0.605)		
**Origin**	**B–ALL**	17	101	19
	**T–ALL**	8	16	4
**Chi square (*p*-value)**		4.91 (0.086)		

## Data Availability

Not applicable.

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
