# Peer review of "Association of TNF–α rs1800629 with Adult Acute B-Cell Lymphoblastic Leukemia"

_genes, 2022, doi:10.3390/genes13071237_

Round 1
Reviewer 1 Report
This study evaluated the frequency of TNF–α rs1800629 polymorphism in adult patients with acute lymphoblastic leukemia.
Major point: Checking the plasma levels of TNF–α of these patients would make the functional connection. Correlation between the polymorphism and TNF–α level is valuable to evaluate whether the TNF–α rs1800629 is a significant risk factor.
Author Response
The paper is well written with a clear focus on the research framework. The introduction is very informative with enough information for the reader about research. The materials and methods are satisfactory and gives enough information for the repeating experiments. However, in my opinion the results and discussion should look shallow and better explained not just to give tables and one or two sentences the authors should at least give their opinion about it.
Authors’ response: The authors believe that the findings are explained with brevity, as brevity is beauty in research writing, with the help of available relevant literature. If the honorable reviewer directs to re-write the findings given in the table in text format, it could be done easily and the authors may do the same (if directed by the reviewer and Editor in Chief).
The conclusion should be also modified because the purpose of the conclusion is to summarize all the obtained data in one paragraph and make easier to reader to understand the research.
Authors’ response: Conclusion has been revised and written in better shape as per reviewer’s instructions.

Reviewer 2 Report
The manuscript of Ezeldine K Abdalhabib et al reports a study that evaluates the frequency of TNF–α rs1800629 (–308 G>A) polymorphism in newly diagnosed adult patients with acute lymphoblastic leukemia (ALL) and its correlation with age at diagnosis, gender and subtype of ALL.
In my opinion the manuscript has enough noveltyn. However, for greater robustness to the results it would be appropriate:
- add some analytical gel that highlights the correct extraction of DNA from the blood
- add some analytical gel that highlights the good amplification of the DNA of patients with the globin gene as a reference
Author Response
The manuscript of Ezeldine K Abdalhabib et al reports a study that evaluates the frequency of TNF–α rs1800629 (–308 G>A) polymorphism in newly diagnosed adult patients with acute lymphoblastic leukemia (ALL) and its correlation with age at diagnosis, gender and subtype of ALL.
In my opinion the manuscript has enough novelty. However, for greater robustness to the results it would be appropriate:
- add some analytical gel that highlights the correct extraction of DNA from the blood
- add some analytical gel that highlights the good amplification of the DNA of patients with the globin gene as a reference
Authors’ response: Added as per directives

Reviewer 3 Report
the paper is well written with a clear focus on the research framework. the introduction is very informative with enough information for the reader about research. the materials and methods are satisfactory and gives enough information for the repeating experiments. however in my opinion the results and discussion should look shallow and better explained not just to give tables and one or two sentences the authors should at least give their opinion about it.
the conclusion should be also modified because the purpose of the conclusion is to summarize all the obtained data in one paragraph and make easier to reader to understand the research.
Author Response

(The authors gave the same response as above.)

Round 2
Reviewer 1 Report
My question was not addressed.
Author Response
Comments and Suggestions for Authors
This study evaluated the frequency of TNF–α rs1800629 polymorphism in adult patients with acute lymphoblastic leukemia.
Major point: Checking the plasma levels of TNF–α of these patients would make the functional connection. Correlation between the polymorphism and TNF–α level is valuable to evaluate whether the TNF–α rs1800629 is a significant risk factor.
Authors’ response:
The authors agree with the plasma levels of TNF–α may provide an insight of a risk factor of adult ALL, however, as this was not the main aim of the study and the authors haven’t stored the plasma for this analysis, this is impossible to carry out to estimate the plasma TNF–α levels.
Secondly, the role of plasma TNF–α levels has been evaluated in other studies including Reference number 10, 27 and 28.
Thirdly, financial constraints are also a major reason of single parameter analysis.

Reviewer 3 Report
the implemented changes improved the paper quality and in my opinion, the article fulfilled the minimum requirements for publication in the journal
Author Response
Comments and Suggestions for Authors
The implemented changes improved the paper quality and in my opinion, the article fulfilled the minimum requirements for publication in the journal.
Authors' response: No action required